# Comprehensive Chemical Profiling in the Ethanol Extract of *Pluchea indica* Aerial Parts by Liquid Chromatography/Mass Spectrometry Analysis of Its Silica Gel Column Chromatography Fractions

**DOI:** 10.3390/molecules24152784

**Published:** 2019-07-31

**Authors:** Jingya Ruan, Jiejing Yan, Dandan Zheng, Fan Sun, Jianli Wang, Lifeng Han, Yi Zhang, Tao Wang

**Affiliations:** 1Tianjin State Key Laboratory of Modern Chinese Medicine, Tianjin University of Traditional Chinese Medicine, 312 Anshanxi Road, Nankai District, Tianjin 300193, China; 2Tianjin Key Laboratory of TCM Chemistry and Analysis, Institute of Traditional Chinese Medicine, Tianjin University of Traditional Chinese Medicine, 312 Anshanxi Road, Nankai District, Tianjin 300193, China

**Keywords:** *Pluchea indica* Less., orthogonal chromatography, normal-phase chromatography silica column separation, reversed-phase liquid chromatography/mass spectrometry analysis, chemical profiling

## Abstract

*Pluchea indica* Less. is a medicine and food dual-use plant, which belongs to the Pluchea genus, Asteraceae family. Its main constituents are quinic acids, flavonoids, thiophenes, phenolic acids, as well as sesquiterpenes. In order to provide a comprehensive chemical profiling of *P. indica*, an orthogonal chromatography combining reverse-phase chromatography BEHC18 column with a normal-phase chromatography silica column as the separation system and a ESI-Q-Orbitrap MS as the detector in both positive and negative ion modes were used. According to the retention time (*t_R_*) and the exact mass-to-charge ratio (*m/z*), 67 compounds were unambiguously identified by comparing to the standard references. Moreover, 47 compounds were tentatively speculated on the basis of the rules of MS/MS fragmentation pattern and chromatographic elution order generalized from the above-mentioned reference standards. Among them, 10 of them were potentially novel.

## 1. Introduction

*Pluchea indica* Less. belongs to the Asteraceae family, and is a kind of medicine and food dual-use plant which mainly distributes in the tropical and subtropical regions. As a kind of amphibious woody semi-mangrove plant, it not only survives in the intertidal zone and becomes a dominant species on the beach, but also naturally breeds in the terrestrial environment. *P. indica* plays a significant role in maintaining ecological balance of coastal areas in the southeast of Asia. Its outstanding effect in softening hardness and dissipating binds has been discovered and utilized by people of the Zhuang nationality in Guangxi for several decades [1]. In vitro cytotoxicity test results indicated that the ethanol extract of *P. indica* leaf exhibited wound healing activity, owing to its ability to improve the cell viability of oral mucosal cell line and accelerate wound closure [2], which suggested that *P. indica* has no toxicity. Next, *P. indica* is not only one of the most famous food ingredients in China [3] but also used as a table salad in Malaysia [4].

Meanwhile, it was also reported that *P. indica* ethanol extract possessed significant inhibitory effects on not only adipocytes lipids and carbohydrate accumulation but also pancreatic lipase activity, which provided evidence for its antiobesity activity [5]. Chemical investigation results suggested the main constituents in this plant are quinic acids, thiophenes, flavonoids, and other phenolic acids [6,7,8]. Among them, quinic acids are the most common dietary polyphenolic compounds, which are widely found in tea, coffee and other foods. It has been reported to exhibit multiple bioactivities such as anti-inflammation and hepatoprotection [4]. What is more, owing to its high content of phenolic compounds, its free radical scavenging activity has been investigated and certified yet, which suggested that *P. indica* displayed antioxidation effect [9,10]. Various thiophenes have been considered to be “generally recognized as safe” (GRAS) by the Flavor and Extract Manufacturers Association (FEMA), and re-evaluated to be safe as a kind of food flavor ingredient according to scientific data [11]. What is more, the usage of the phenolic acids in dietary has also been popularized all over the world [12].

The application of *P. indica* in food additives is more and more extensive. However, during the chemical researches in our lab, we found a large amount of pigment existed in the plant made it difficult to accomplish the comprehensive characterization of chemical composition. Meanwhile, the analysis directly acomplished by LC-MS usually limited the characterization fewer than 50 compounds [13,14]. A strategy that could improve the chromatographic peak capacity should be considered [15]. As we known, the constituents in *P. Indica* are varieties, and ideal separation effect is often hardly obtained by conventional reversed-phase chromatography [16]. Then, preliminary fractionation for its crude extracts by normal-phase silica gel column was utilized before reversed-phase chromatography analysis.

According to our previous study [7,8], the main constituents in *P. indica* are known to be quinic acids, flavonoids, thiophenes, phenolic acids, and sesquiterpenes. And in the process of LC-MS analysis, we found that in the negative ion mode, fragments from [M – H]^–^ ions could provide more comprehensive information for quinic acids, flavonoids, and phenolic acids, while thiophenes and sesquiterpenes were more inclined to be detected by positive ion mode. Thus, in this study, the ESI-Q-Orbitrap MS was used as the detector in both positive and negative ion modes.

This paper aimed to use a LC-MS analysis method by an orthogonal chromatography (OC) integrating reversed-phase chromatography BEHC18 column with normal-phase chromatography silica column to achieve the characterization and identification of *P. indica* comprehensively. Six fractions of 95% eluates of the aerial parts of *P. indica* (PIE1–PIE6) were subsequently analyzed by LC-MS on an ESI-Q-Orbitrap MS in both negative and positive ion modes. According to the retention time (*t_R_*) and the exact mass-to-charge ratio (*m*/*z*), 67 compounds were unambiguously identified by comparing to the standard references. Meanwhile, the rules of MS/MS fragmentation pattern and chromatographic elution order have been generalized by using the reference standards, 47 compounds were tentatively speculated, and 10 of them were potential new ones.

## 2. Experimental

### 2.1. Materials

The aerial parts of *P. indica* were collected from Hepu City, Guangxi Province, China, and identified by Dr. Songji Wei (Zhuang Medical College of Guangxi University of Chinese Medicine, Nanning, China). The voucher specimen was deposited at the institute of traditional Chinese Medicine of Tianjin University of TCM.

Sixty-seven compounds, including 20 quinic acids (Appendix A), 19 phenolic acids (Appendix A), 4 thiophenes (Appendix A), 13 flavonoids (Appendix A), and 11 others (Appendix A), were used for reference standards. Their purities were >98%. Among them, 60 (listed following) were prepared by ourselves, and 7 were purchased.

Among the 20 quinic acids, neochlorogenic acid (5-caffeoylquinic acid), methyl 3-*O*-caffeoyl quinate, and 1,5-di-*O*-caffeoyl quinic acid were purchased from Must Company (Chengdu, China); and 17 of them, chlorogenic acid (3-caffeoylquinic acid), 4-caffeoylquinic acid, 1,3-di-*O*-caffeoyl quinic acid, 1,4-di-*O-*caffeoyl quinic acid, 3,4-di-*O*-caffeoyl quinic acid, 3,5-di-*O*-caffeoyl quinic acid, 4,5-di-*O*-caffeoyl quinic acid, methyl 3,4-di-*O*-caffeoyl quinate, methyl 3,5-di-*O*-caffeoyl quinate, methyl 4,5-di-*O*-caffeoyl quinate, ethyl 3,4-di-*O*-caffeoyl quinate, ethyl 3,5-di-*O*-caffeoyl quinate, 1,3,4-tri-*O*-caffeoyl quinic acid, 1,3,5-tri-*O*-caffeoyl quinic acid, 3,4,5-tri-*O*-caffeoyl quinic acid, methyl 3,4,5-tri-*O*-caffeoyl quinate, and 1,3,4,5-tetra-*O*-caffeoyl quinic acid were isolated from the aerial parts of *P. indica* and *Chrysanthemum morifolium* [17].

The phenolic acids, 3,4-dihydroxy benzoic acid, 3-methoxy-4-hydroxybenzoic acid, *p*-hydroxybenzoic acid, and *trans*-ferulic acid were purchased from Must Company (Chengdu, China). Other 15 ones, 3,4-dihydroxy benzaldehyde, vanillin, 3,4-dihydroxy-5-methoxybenzaldehyde, syringicaldehyde, dibutylphthalate, ethyl caffeate, 2,3-dihydroxy-1-(4-hydroxy-3-methoxyphenyl)-propan-1-one, *trans*-coniferyl aldehyde, esculetin, *threo*-2,3-bis(4-hydroxy-3-methoxyphenyl)-3-ethoxypropan-1-ol, *erythro*-2,3-bis-(4-hydroxy-3-methoxyphenyl)-3-ethoxypropan-1-ol, (+)-isolariciresinol, (–)-(7*S*,7'*S*,8*R*,8'*R*)-4,4'-dihydroxy-3,3',5,5'-pentamethoxy-7,9':7',9-diepoxylignane, (+)-9'-isovaleryllariciresinol, and *trans*-caffeic acid were obtained from the aerial parts of *P. indica* [7], *C. morifolium* [18], *D. hypoglauca*, and *L. japonicas*.

The thiophenes, (3''*R*)-pluthiophenol, (3''*R*)-pluthiophenol-4''-acetate, 3''-ethoxyl-(3''*S*)-pluthiophenol, and 3''-ethoxyl-(3''*S*)-pluthiophenol-4''-acetate (Appendix A) were isolated from the aerial parts of *P. indica* [7].

Thirteen flavonoids (Appendix A) including quercetin, quercetin-3-*O*-β-d-glucopyranoside, quercetin-3-*O*-β-d-galactopyranoside, isorhamnetin, 5,6,4'-trihydroxy-3,7-dimethoxyflavone, centaureidin, chrysosplenol C, casticin, cynaroside, luteolin, kaempferol 3-*O*-β-d-glucopyranoside (astragalin), 5,7,3',4'-tetrahydroxy-3-methoxyflavonol-3'-*O*-β-d-glucopyranoside, and kaempferol were isolated from the aerial parts of *P. indica* [8], *C. morifolium* [17], and *L. leontopodioides* by us.

Eleven other compounds (Appendix A), caryolane-1,9β-diol, (8*R*,9*R*)-isocaryolane-8,9-diol, clovane-2α,9β-diol, valenc-1(10)-ene-8,11-diol, fraxinellone, stigmasterol, methyl 10-oxoundecanoate, triethyl citrate, 9,12,13-trihydroxyoctadeca-10(*E*),15(*Z*)-dienoic acid, and pinellic acid together with adenosine were obtained from *P. indica* aerial parts [7].

HPLC grade Acetonitrile (Thermo-fisher, Waltham, MA, USA) and ultrapure water prepared with a Milli-Q purification system (Merck Millipore, Darmstadt, Hesse-Darmstadt, Germany) were used for LC-MS analysis. Analytical-grade ethanol (EtOH), chloroform (CHCl_3_), and methanol (MeOH) (Haiguang Chemical Reagent Factory, Tianjin, China), as well as silica gel (48–75 µm, Haiyang Chemical Reagent Factory, Qingdao, China) were used for the preparation of *P. indica* 70% EtOH extract (PI) and silica gel fractionation of 95% EtOH eluate (PIEs) test solutions.

### 2.2. Sample Preparation

#### 2.2.1. Preparation of Standard Solutions

Standard test solutions of above mentioned standard references were prepared in MeOH at a final concentration of ~100 ng/mL. All stock solutions were stored at 4 °C in darkness and brought to room temperature before use.

#### 2.2.2. Preparation of *P. indica* 70% EtOH Extract Test Solutions

An aliquot of 100 g dried aerial parts of *P. indica* was extracted under reflux in 800, 600, and 600 mL 70% ethanol (*v*/*v*) for 3, 2, and 2 h, respectively. The combined extracts were filtered with 0.22 µm microporous membrane to obtained PI stock solutions. PI stock solutions were stored at 4 °C in darkness and brought to room temperature before use.

#### 2.2.3. Preparation of Silica Gel Fractionation of 95% EtOH Eluate of *P. indica* Test Solutions

The above-mentioned PI stock solutions were evaporated to dryness under reduced pressure to obtain PI extract (20.55 g). The PI extract was dissolved in 800 mL of water and separated by D101 macroporous adsorption resin column (H_2_O → 95% EtOH) to obtain H_2_O eluate (15.45 g) and 95% EtOH eluate (PIE, 5.0 g), respectively. PIE (5.0 g) were subjected to normal-phase silica gel (50.0 g) column (CHCl_3_-MeOH (100:1 → 100:5, *v*/*v*) → CHCl_3_-MeOH-H_2_O (10:3:1 → 7:3:1 → 6:4:1 → 5:5:1, *v*/*v*/*v*, lower layer) → MeOH), six fractions (PIE1–PIE6) were obtained. PIE1–PIE6 were evaporated to dryness under reduced pressure, and then dissolved with MeOH to get six test stock solutions at a final concentration of 10 mg/mL. PIEs stock solutions were stored at 4 °C in darkness and brought to room temperature before use.

### 2.3. UHPLC

A Thermo UltiMate 3000 UHPLC instrument (Thermo, Waltham, MA, USA) equipped with a quaternary pump and an autosampler were used to accomplish the analysis. Samples were separated on a Waters ACQUITY UPLC® BEH C18 (2.1 × 100 mm, 1.7 μm, Milford, MA, USA) using a mobile phase composed of H_2_O (A) and CH_3_CN with 0.1% CH_3_COOH (B) in the following gradient program; 0–4 min, 1–4% B; 4–18 min, 4% B; 18–20 min, 4–13% B; 20–24 min, 13% B; 24–35 min, 13–15% B; 35–37 min, 15–24% B; 37–45 min, 24–30% B; 45–46 min, 30–55% B; 46–52 min, 55% B; 52–53 min, 55–95% B; 53–58 min, 95% B. An equilibration of 3 min was used between successive injections. The flow rate was 0.4 mL/min, and column temperature was 35 °C. An aliquot of 3 μL of each sample was injected for analysis.

### 2.4. ESI-Q-Orbitrap MS and Automatic Components Extraction

For tandem mass spectrometry analysis, a Thermo ESI-Q-Orbitrap MS mass spectrometer was connected to the UltiMate 3000 UHPLC instrument via ESI interface. Ultrahigh purity nitrogen (N_2_) was used as the collision gas and the sheath/auxiliary gas. The ESI source parameters were set as follows: ion spray voltage 3.2 kV, capillary temperature 350 °C, ion source heater temperature 300 °C, sheath gas (N_2_) 40 arbitrary units, auxiliary gas (N_2_) 10 arbitrary units, and a normalized collision energy (NCE) of 35 V was used. The Orbitrap analyzer scanned the mass range from *m/z* 100 to 1500. Monitoring time was 0–58 min. Detection was obtained by full mass-dd mass mode. The MS data were recorded in both profile and centroid formats. Data recording and processing were performed using the Xcalibur 4.0 software (Thermo Fisher Scientific, Inc., Waltham, MA, USA). The accuracy error threshold was fixed at 5 ppm.

Software-aided, automatic background subtraction and components extraction techniques were used to generate a peak list containing all the components profiled from the aerial part of *P. indica*. Sieve v2.2 SP2 (Thermo Fisher Scientific) was used for the automatic components extraction: time range: 1–58 min; BP minimum count: 10,000; BP minimum scans: 5; Background SN: 3; MZ Step: 10; and Frame, >1.

## 3. Results and Discussion

OC-MS technology integrated the large peak capacity of OC with the high detected sensitivity of MS, which could accomplish more comprehensive chemical profiling. Through comparing the separation effects of four kinds of high performance liquid chromatography columns (HSS C18, T3, CSH FP, and BEH C18 columns) with different separation mechanisms for PI, BEH C18 column was found to have better separation ability for most of the peaks in PI (Appendix A). Then, it was chosen to carry out the composition analysis of *P. indica*. The comprehensive and accurate chemical composition profiling of *P. indica* was further accomplished by the OC-MS combined normal-phase silica gel column with reverse-phase BEH C18 column (Figure 1 and Figure 2). As a result, the peak capacity and detection sensitivity were improved significantly, which is discussed in detail as follows.

### 3.1. Superiority of OC-MS than Directly LC-MS in the Chemical Composition Profiling of P. indica 70% EtOH Extract

#### 3.1.1. Increasing of the Peak Capacity by OC-MS Determination

OC-MS, combing the advantages of the large peak capacity of OC and high sensitivity of mass spectrometry, had a wide range of applications, especially in the systematic chemical composition characterization of Traditional Chinese Medicine [15].

The sieve statistical results indicated that OC-MS possessed much more peaks than directly LC-MS (PI: 234 peaks detected; PIEs: 735 peaks detected).

Taking the ion peak at 40.13 min as an example, when extracting it from the total ion chromatogram of PI, only the ion at *m*/*z* 677.15147 corresponding to 3,4,5-tricaffeoyl quinic acid could be observed obviously (Appendix A). However, two ion peaks of belonging to 3,5-dicaffeylquinine ethyl acetate (*m*/*z* 543.15136) and 3,4,5-tricaffeoylquinic acid (*m*/*z* 677.15126) (Appendix A) were extracted from the ion peak at 40.13 min of PIE4. And both of them appeared with extremely strong intensity. This result suggested that OC-MS technique could increase peak capacity of the chromatography greatly.

#### 3.1.2. Increasing of the Detection Sensitivity by OC-MS Determination

Ion peaks of *m/z* 677.15119 were extracted from the total ion chromatogram of PI and PIE6 (Appendix A). It was found that the peak area ratios of 1,3,5-tricaffeoyl quinic acid (36.63 min), 1,3,4-tricaffeoylquinic acid (38.13 min), and 1,4,5-tricaffeoylquinic acid (39.86 min) to 3,4,5-tricaffeoylquinic acid (40.11 min) in PIE5 (0.15, 0.11, and 0.09, respectively) were significantly higher than those in PI (0.03, 0.03, and 0.01, respectively). The above-mentioned results indicated that OC-MS technique could improve detection sensitivity.

### 3.2. Structural Elucidation of Compounds from Silica Gel Fractionation of 95% EtOH Eluate of P. indica by OC-MS

#### 3.2.1. Structural Elucidation of Quinic Acids and Their Derivatives

Quinic acids and their derivatives are one kind of the most characteristic components in Asteraceae family plants [19,20,21]. There are four hydroxyl groups substituted at 1-, 3-, 4-, and 5-position, as well as one carboxyl at 1-position of quinic acid, respectively, which means that the quinic acids derivatives can be formed at any one of above mentioned position or functional group theoretically. In fact, 1-, 3-, 4-, and 5-hydroxyl can dehydrate with caffeic acid, ferulic acid, coumaric acid, and 3,4,5-trihydroxycinnamic acid to form acyl-substituted quinic acids. What is more, the 1-carboxyl is easily to form methyl and ethyl esters.

##### Structural Elucidation of Mono-acyl Substituted Quinic Acids and Their Derivatives

Peaks **4** (*m/z* 353.08710 [M − H]^–^), **11** (*m/z* 353.08716 [M – H]^–^), and **16** (*m/z* 353.08699 [M – H]^–^) were unambiguously confirmed to be 5-caffeoylquinic acid (5-CQA), 3-CQA, and 4-CQA, respectively, by comparing to the standard references obtained from *C. morifolium* [17]. As a result, the general MS/MS cleavage pattern of CQAs was summarized, which mainly included ester bond cleavage between caffeoyl and quinic acid groups, quinic acid dehydration, along with caffeoyl decarbonylation. Then, characteristic fragment ions at *m/z* 179.03389 [caffeoyl − H]^−^, 161.02332 [caffeoyl – H − H_2_O]^−^, 135.04406 [caffeoyl – H − CO_2_]^−^, 133.02841 [caffeoyl – H − H_2_O − CO]^−^ for caffeoyl group and *m/z* 191.05501 [quinic acid − H]^−^, 173.04445 [quinic acid – H − H_2_O]^−^ for quinic acid group were displayed in their mass spectra (Figure 3).

Comparing the MS/MS fragmentation characteristics and relative ion abundance of CQAs, it was found that the fragment ion type and ion intensity had a great correlation with the substitution position of caffeoyl group: in the MS/MS spectrum for 3-CQA, the characteristic fragment ion at *m/z* 191.05501 derived from quinic acid aglycone was the base peak; however, the ion peaks at *m/z* 179.03389, 161.02332, 135.04406, and 133.02841 for caffeoyl group and *m/z* 173.04445 for quinic acid group were very weak; in the MS/MS spectrum for 4-CQA, the ion peaks at *m/z* 191.05501 and 173.04445 for quinic acid group and 179.03389 and 135.04406 for caffeoyl group were stronger, and the base peak was *m/z* 173.04445 from quinic acid aglycone; meanwhile, in the MS/MS spectrum for 5-CQA, not only the characteristic fragment ions of *m/z* 191.05501 for quinic acid nucleus existed as base peak, but also the strong caffeoyl characterization fragment ions at *m/z* 179.03389 and 135.04406 displayed (Appendix A). This phenomenon has also been observed in the study of chlorogenic acids in Kuding Tea [22]. This feature might be caused by the axial conformation of 5-caffeoyl group, which made it easy for the cleavage of 5-caffeoyl group from the quinic acid nucleus to form relative stable fragmentation ions of both quinic acid and caffeoyl group.

Moreover, the chromatographic elution order [retention time (*t*_R_)] of CQAs in the CH_3_CN-H_2_O system was denoted as 5-CQA (peak **4**, *t*_R_ 4.97 min) < 3-CQA (peak **11**, *t*_R_ 8.20 min) < 4-CQA (peak **16**, *t*_R_ 10.69 min).

According to the reference [23], the substituted group of quinic acid aglycone could also be feruloyl, and coumaroyl except for caffeoyl; mono-feruloyl quinic acid (FQAs) and mono-coumaroyl quinic acid (CoQAs) were also formed. Moreover, the possibility of 3,4,5-trihydroxycinnamoyl substitution (another kind of caffeoyl derivatives) has also been considered, which could form mono-3,4,5-trihydroxycinnamoyl quinic acid (TQAs). The characteristic ions of caffeoyl, feruloyl, coumaroyl, and 3,4,5-trihydroxycinnamoyl were considered to be at *m/z* 179.03389 ([caffeoyl − H]^−^), 193.04954 ([feruloyl − H]^−^), 163.03897 ([coumaroyl − H]^−^), and 195.02880 ([3,4,5-trihydroxycinnamoyl − H]^−^), respectively, which could be used to distinguish the type of substituted group.

In the MS/MS fragmentation spectra of peaks **3** (*m/z* 369.08246 [M – H]^–^), **9** (*m/z* 369.08266 [M – H]^–^), and **15** (*m/z* 369.08233 [M – H]^–^) (Appendix A), there were noticeable *m/z* 195.02880 (C_9_H_7_O_5_^−^) ions, which indicated that they were substituted by 3,4,5-trihydroxycinnamoyl group. The base peak of the MS/MS fragmentation spectra for peaks **3**, **9**, and **15** was *m/z* 191.05483, 133.02803, and 173.04421, respectively, and there were relatively strong ion at *m/z* 133.02803 in that of peak **3**. Thus, based on the above rules of MS/MS cleavage patterns and chromatographic elution order generalized by 5-CQA (peak **4**), 3-CQA (peak **11**), and 4-CQA (peak **16**), they were tentatively deduced to be 5-TQA (peak **3**), 3-TQA (peak **9**), and 4-TQA (peak **15**), respectively. They were all possible new ones. Using the same method, peaks **18** (*m/z* 337.09254 [M – H]^–^) and **20** (*m/z* 337.09268 [M – H]^–^) were tentatively identified as 3-CoQA and 4-CoQA, respectively (Appendix A).

Peak **29** (*m/z* 367.10270 [M – H]^–^) was unambiguously identified as 3-CQM by comparing with reference standard. Though the molecular of peaks **21** (*m/z* 367.10266 [M – H]^–^), **22** (*m/z* 367.10208 [M – H]^–^), **24** (*m/z* 367.10254 [M – H]^–^), and **28** (*m/z* 367.10260 [M – H]^–^) were similar with that of peak **29**; the appearance of *m/z* 193.04967 [feruoyl − H]^−^, 193.04965 [feruoyl − H]^−^ and 193.04939 [feruoyl − H]^−^ in the MS/MS spectra of peaks **21**, **22**, and **24**, respectively, indicated that these three compounds were substituted by feruoyl group. According to the chromatographic elution order summarized above, peaks **21**, **22**, and **24** were tentatively identified as 5-FQA, 3-FQA, and 4-FQA, respectively. In the MS/MS spectrum of peak **28**, the ions at *m/z* 193.04954 [feruoyl − H]^−^ was absent, while *m/z* 179.03380 [caffeoyl − H]^−^, 161.02313 [caffeoyl – H − H_2_O]^−^ appeared, then it was deduced to be CQM. According to the chromatographic elution order of CQAs (*t*_R_: 5-CQA < 3-CQA < 4-CQA), and combining the occurrence probability of 1-CQM and 5-CQM, peak **28** was tentatively deduced to be 5-CQM. And it was found to be a new compound (Appendix A).

##### Structural Elucidation of Multi-acyl Substituted Quinic Acids and Their Derivatives

Peaks **35** (*m/z* 515.11761 [M – H]^–^), **37** (*m/z* 515.11694 [M – H]^–^), **38** (*m/z* 515.11797 [M – H]^–^), **39** (*m/z* 515.11713 [M – H]^–^), **40** (*m/z* 515.11950 [M – H]^–^), and **45** (*m/z* 515.11523 [M – H]^–^) were definitively identified as 1,4-dicaffeoylquinic acid (1,4-diCQA), 1,5-diCQA, 1,3-diCQA, 4,5-diCQA, 3,5-diCQA, and 3,4-diCQA, respectively, by comparing to the standard references obtained from *C. morifolium* and *P. indica*. According to the MS/MS spectra of the six diCQAs, the general MS/MS fragmentation pattern diCQAs was summed up as following: the ion peak at *m/z* 173.04434 [M – H – 2C_9_H_6_O_3_ – H_2_O]^–^, 173.04420 [M – H – 2C_9_H_6_O_3_ – H_2_O]^–^, and 173.04430 [M – H – 2C_9_H_6_O_3_ – H_2_O]^–^ derived from quinic acid aglycone was found to be the base peak in the MS/MS spectra of **35** (1,4-diCQA), **39** (4,5-diCQA), and **45** (3,4-diCQA), respectively (Appendix A); the base peak of **37** (1,5-diCQA) was *m/z* 135.04356 [M – H – C_9_H_6_O_3_ – C_7_H_10_O_5_ – CO_2_]^–^ for caffeoyl group (Appendix A); however, for **38** (1,3-diCQA) and **40** (3,5-diCQA), *m/z* 191.05495 [M – H – 2C_9_H_6_O_3_]^–^ and 191.05480 [M – H – 2C_9_H_6_O_3_]^–^ derived from quinic acid aglycone was the base peak, respectively (Appendix A). The above mentioned rules was similar to those of CQAs, which suggested that when C-4 was substituted by caffeoyl (**35**, **39**, **45**), *m/z* 173.04445 was an indicator ion as its strong intensity (Appendix A).

In addition, the chromatographic elution order of diCQAs in the CH_3_CN-H_2_O system was showed as 1,4-diCQA (peak **35**, *t*_R_ 27.11 min) < 1,5-diCQA (peak **37**, *t*_R_ 27.99 min) < 1,3-diCQA (peak **38**, *t*_R_ 28.07 min) < 4,5-diCQA (peak **39**, *t*_R_ 28.68 min) < 3,5-diCQA (peak **40**, *t*_R_ 28.89 min) < 3,4-diCQA (peak **45**, *t*_R_ 34.01 min). Meanwhile, comparing with the *t*_R_ of CQAs, that of diCQAs was longer significantly. On the other hand, *t*_R_ of C-1 substituted diCQAs was shorter than that of other diCQAs.

Using the above-mentioned rules, the peaks **46** (*m/z* 529.13623 [M – H]^–^), **48** (*m/z* 529.13515 [M – H]^–^), **49** (*m/z* 529.13452 [M – H]^–^) were tentatively deduced. The research process was described as following: in the MS/MS spectra of peaks **46**, **48**, and **49**, there were *m/z* 193.04970 [M – H – C_16_H_16_O_8_]^–^, 193.04991 [M – H – C_16_H_16_O_8_]^–^, and 193.04980 [M – H – C_16_H_16_O_8_]^–^ for **46**, **48**, and **49**, respectively, along with 134.03615 [M – H – C_16_H_16_O_8_ – CH_3_ – CO_2_]^–^, 134.03642 [M – H – C_16_H_16_O_8_ – CH_3_ – CO_2_]^–^, and 134.03607 [M – H – C_16_H_16_O_8_ – CH_3_ – CO_2_]^–^ for **46**, **48**, and **49**, respectively (characteristic fragment ions for feruloyl group); as well as *m/z* 179.03404 [M – H − C_17_H_18_O_8_]^−^, 179.03429 [M – H − C_17_H_18_O_8_]^−^, and 179.03400 [M – H − C_17_H_18_O_8_]^−^, respectively (characteristic fragment ion for caffeoyl group). Therefore, it was presumed that they were CFQAs substituted by a mono-caffeoyl group and a mono-feruloyl group in different positions of quinic acid aglycone at the same time. As the characteristic fragment ion *m/z* 173.04445 was not the base peak ion of their MS/MS spectra, the possibility that caffeoyl or feruloyl linked at C-4 of the quinic acid aglycone was excluded. Combining the chromatographic elution order of diCQAs in the CH_3_CN-H_2_O system summarized above, peaks **46**, **48**, and **49** were deduced to be 1-C-5-FQA/5-C-1-FQA, 1-C-3-FQA/3-C-1-FQA, and 3-C-5-FQA/5-C-3-FQA, respectively. Moreover, in the dominant conformation of the six-membered ring quinic acid aglycone, both 1- and 5-OH were at the axial bond, the substitution on C-1 and 5 were more likely to cleavage from the aglycone in the mass spectrometry than they were at other positions. Thus according to the stability of fragment ions (relative abundance) after substituent cleavage, the substitution position could be preliminary identified. For example, in peak **49**, the fragment ion intensity of *m/z* 367.10309 [M – H – C_9_H_6_O_3_]^–^ (derived from the cleavage of one caffeoyl group 162 Da) was high, which suggested that the caffeoyl group was linked to the C-5 of quinic acid aglycone. Therefore, it was finally determined that the peak **49** was 3-feruloyl-5-caffeoyl-quinic acid. Using a similar method, peak **48** was speculated to be 1-caffeoyl-3-feruloyl-quinic acid (Appendix A).

Peaks **51** (*m/z* 529.13409 [M – H]^–^), **55** (*m/z* 529.13403 [M – H]^–^), **61** (*m/z* 529.13367 [M – H]^–^) (Appendix A), **66** (*m/z* 543.15027 [M – H]^–^), and **74** (*m/z* 543.15080 [M – H]^–^) (Appendix A) were identified as methyl 4,5-dicaffeoylquinate (4,5-diCQM), 3,5-diCQM, 3,4-diCQM, ethyl 4,5-dicaffeoylquinate (3,5-diCQE), and 3,4-diCQE, respectively, by comparing with reference standards isolated from *P. indica*. Their MS/MS fragment patterns were similar to those of CQMs mentioned above. In detail, in the MS/MS spectra of 4,5-diCQM (peak **51**), 3,5-diCQM (peak **55**), and 3,5-diCQE (peak **66**), the base peak ion was *m/z* 161.02306 [529.13515 – C_9_H_6_O_3_ – C_7_H_12_O_6_ – CH_2_]^–^, 161.02304 [529.13515 – C_9_H_6_O_3_ – C_7_H_12_O_6_ – CH_2_]^–^, and 161.02316 [543.15080 – C_9_H_6_O_3_ – C_7_H_12_O_6_ – C_2_H_4_]^–^, respectively. What is more, and the abundance of *m/z* 135.04356 [529.13515 – C_9_H_6_O_3_ – C_7_H_10_O_5_ – CH_2_ – CO_2_]^–^ and 133.02808 [543.15080 – C_9_H_6_O_3_ – C_7_H_12_O_6_ – C_2_H_4_ – CO]^–^ was also strong in the MS/MS spectra of **55** and **66**, respectively. While, *m/z* 135.04364 [M – H – C_9_H_6_O_3_ – C_7_H_10_O_5_ – CH_2_ – CO_2_]^–^ and 135.04370 [M – H – C_9_H_6_O_3_ – C_7_H_10_O_5_ – C_2_H_4_ – CO_2_]^–^ was the base peak for the MS/MS spectra of 3,4-diCQM (peak **61**) and 3,4-diCQE (peak **74**), respectively. These indicated when 3- and/or 5-OH of CQMs/CQEs were substituted by caffeoyl; the base peak ion was at *m/z* 135.04406 or 161.02332. On the other hand, their chromatographic elution order (4,5-diCQM (peak **51**, *t*_R_ 37.80 min) < 3,5-diCQM (peak **55**, *t*_R_ 38.39 min) < 3,4-diCQM (peak **61**, *t*_R_ 39.49 min); 3,5-diCQE (peak **66**, *t*_R_ 40.16 min) < 3,4-diCQE (peak **74**, *t*_R_ 41.61 min)) was consistent with that of di-CQAs. As a result, peak **50** (*m/z* 529.13515 [M – H]^–^) was tentatively identified as 1,4-diCQM according to the base peak ions *m/z* 191.05522 [M – H – 2C_9_H_6_O_3_ – CH_2_]^–^ and its *t_R_* (peak **50**, 37.55 min). According to the base peak ions *m/z* 161.02313 [M – H – C_9_H_6_O_3_ – C_2_H_4_ – C_7_H_12_O_6_]^–^ and the *t_R_* (peak **62**, 39.59 min) (Appendix A), peak **62** (*m/z* 543.15080 [M – H]^–^) was tentatively identified as 1,5-diCQE or 4,5-diCQE.

It was visible from the above derivation processes that CQFAs were confusing with diCQMs as they possessed the same molecular. The key to distinguish them was the presence of characteristic fragment ions at *m/z* 193.04954 and 134.03623 for the feruloyl group, or not [23].

Peaks **47** (*m/z* 677.15277 [M – H]^–^), **53** (*m/z* 677.15070 [M – H]^–^), and **65** (*m/z* 677.15270 [M – H]^–^) (Appendix A) were unambiguously identified as 1,3,5-tricaffeoyl quinic acid (1,3,5-triCQA), 1,3,4-triCQA, 3,4,5-triCQA, respectively, by comparing with reference standards obtained from *P. indica*. As we have mentioned above, the quinic acid aglycone could be substituted by multiple acyls at its 1-, 3-, 4-, and 5-hydroxyl, thus the possibilities of triCQAs included 1,3,5-triCQA, 1,3,4-triCQA, 1,4,5-triCQA, and 3,4,5-triCQA. Then, another triCQA (peak **63**) should be deduced to be 1,4,5-triCQA.

Peak **76** (*m/z* 839.18289 [M – H]^–^) (Appendix A), was definitely confirmed to be 1,3,4,5-tetracaffeoyl quinic acid (1,3,4,5-tetraCQA) by comparing with reference standards. Peak **92** (*m/z* 867.21393 [M – H]^–^) (Appendix A) was tentatively identified as ethyl 1,3,4,5-tetracaffeoyl quinate (1,3,4,5-tetraCQE), because of the lack of feruloyl group characteristic fragment ions as well as the similarity of its MS/MS spectra with that of 1,3,4,5-tetraCQM (peak **76**) (Appendix A). What is more, peak **92** was possible to be a new compound.

Peak **91** (*m/z* 691.16684 [M – H]^–^) (Appendix A) was definitely identified as methyl 3,4,5-tricaffeoyl quinate (3,4,5-triCQM) by comparing with reference standards. In the MS/MS spectrum of peak **80** (*m/z* 691.16754 [M – H]^–^) (Appendix A), feruloyl group characteristic fragment ions *m/z* 193.04954 and 134.03623 were lacking, while *m/z* 179.03 [M – H − 2C_9_H_6_O_3_ − C_7_H_10_O_5_ − CH_2_]^−^ for caffeoyl group was observed, thus it was speculated to be a triCQM. Because of the lack of samples for summary of MS fragment pathway and retention time rules, peak **80** was tentatively deduced to be 1,3,4-triCQM or 1,3,5-triCQM or 1,4,5-triCQM. In addition, owing to the lack of feruloyl group characteristic fragment ions, and its MS/MS spectrum was similar to that of 3,4,5-triCQM (peak **91**), peak **98** (*m/z* 705.18036 [M – H]^–^) (Appendix A) was tentatively identified as ethyl 3,4,5-tricaffeoyl quinate (3,4,5-triCQE). Moreover, peak **98** is potentially novel.

Meanwhile using a similar manner mentioned above, the structure types of the remaining 13 acyl-substituted quinic acids and their derivatives were tentatively proposed (Table 1).

#### 3.2.2. Structural Elucidation of Phenolic Acids

Peaks **5**, **6**, **10**, **13**, **14**, **19**, **23**, **33**, **34**, **52**, **59**, **74**, **81**, **109**, **112**, and **113** were accurately identified by comparison with reference standards (Table 1, Appendix A).

Peaks **1** and **7** were unambiguously identified as 3,4-dihydroxy benzoic acid and 3,4-dihydroxybenzaldehyde by comparison with reference standards isolated from *P. indica* [7], respectively. Figure 4 showed the MS/MS fragmentation pattern of benzoic acid (peak **1**) and benzaldehyde (peak **7**): the alpha bond cleavage of carbonyl was susceptible to occur in benzoic acid and benzaldehyde, and then generated a fragment ion peak at *m/z* 109.02793 and 109.02798 by removing 44 Da and 28 Da, respectively. Thus, the two types of compounds could be distinguished by the mass lost during this process (Appendix A). Peak **12** was definitely identified as 3-methoxy-4-hydroxybenzoic acid by comparison with reference standard. Besides the rule mentioned above, it was also prone to remove one methyl radical, resulting in a strong fragment ion peak at *m/z* 152.01041 [M – H – CH_3_]^–^; at the same time, it could remove one CO_2_ to form a fragment ion peak of *m/z* 123.04406 [M – H – CO_2_]^–^. Continually, it could remove one CO_2_ or methyl radical to form fragment ion peak of *m/z* 108.02058 [M – H – CH_3_ – CO_2_]^–^/[M – H – CO_2_ – CH_3_]^–^, respectively (Figure 4 and Appendix A).

Peak **8** (*m/z* 167.03381 [M – H]^–^) (Appendix A), which has the molecular ion C_8_H_7_O_4_^–^ in its MS/MS spectrum, had fragment ion peaks of *m/z* 152.01022 [M – H – CH_3_]^–^ and 108.02016 [M – H – CH_3_ – CO_2_]^–^, suggesting that one methyl radical (15 Da) and CO_2_ (44 Da) were cleaved from the structure. The existence of 3,4-dihydroxybenzoic acid (**1**) in this plant has been proved above, combining the biosynthetic pathway, peak **8** was tentatively presumed to be methyl 3,4-dihydroxybenzoate.

In the MS/MS spectrum of peak **17** (*m/z* 137.02303, C_7_H_5_O_3_^–^) (Appendix A), one CO_2_ (44 Da) was cleaved according to the base peak ion of *m/z* 93.03311 [M – H – CO_2_]^–^. Therefore, it was deduced that there was a carboxyl group directly attached to the benzene ring in the structure. As a result, peak **17** was tentatively speculated to be 2-hydroxybenzoic acid or 3-hydroxybenzoic acid combining with its molecular formula.

#### 3.2.3. Structural Elucidation of Thiophenes

As mentioned above, four thiophenes: (3''*R*)-pluthiophenol (peak **99**, *m/z* 231.04743, *t*_R_ 46.82 min), (3''*R*)-pluthiophenol-4''-acetate (peak **105**, *m/z* 273.05813, *t*_R_ 47.94 min), 3''-ethoxyl-(3''*S*)-pluthiophenol (peak **107**, *m/z* 259.07918, *t*_R_ 49.42 min), and 3''-ethoxyl-(3''*S*)-pluthiophenol-4''-acetate (peak **111**, *m/z* 301.08917, *t*_R_ 53.25 min) have been isolated and identified in our lab [7]. When extracting the above ions from PIE2, four pairs of ion peaks were obtained, the four peaks unseparated were presumed to be (3''*S*)-pluthiophenol (peak **106**, *m/z* 231.04743, *t*_R_ 48.73 min), (3''*S*)-pluthiophenol-4''-acetate (peak **110**, *m/z* 273.05813, *t*_R_ 53.13 min), 3''-ethoxyl-(3''*R*)-pluthiophenol (peak **97**, *m/z* 259.07918, *t*_R_ 46.55 min) and 3''-ethoxyl-(3''*R*)-pluthiophenol-4''-acetate (peak **104**, *m/z* 301.08917, *t*_R_ 47.62 min) (Figure 5). In addition, the fragmentation ions of peaks **110** and **97** were similar to those of peaks **105** and **107** (Table 1), respectively, which preliminary confirmed the above inference. While the MS/MS fragmentation ions of peaks **104** and **106** have not been detected. Summarizing their chromatographic elution order, it was found that in the CH_3_CN-H_2_O system the *t*_R_ of the thiophenes with the same planar structure was related to the configuration: the *t*_R_ of 3''*R*-thiophene was shorter than that of 3''*S*-thiophene. The four thiophenes peaks **97**, **104**, **106**, and **110** could be tentatively identified due to their peak orders, and all of them were possible new compounds.

#### 3.2.4. Structural Elucidation of Other Compounds

In addition, 13 flavonoids (Appendix A) and 11 other compounds (Appendix A) were unambiguously identified by comparison to reference standards obtained and identified by our lab [7,8] (Table 1). At the same time, when taking literatures into consideration, the possible structures of 11 compounds (9 flavonoids and two sesquiterpenes) were tentatively predicted (Table 1).

During this process, an interesting phenomenon has been found that when the 1-position carboxyl of quinic acids were methylated, the base peak of the MS/MS spectra of CQMs were not 191.05501 [quinic acid − H]^−^ or 173.04445 [quinic acid – H − H_2_O]^−^ any more, they became 161.02332 [caffeoyl − H_2_O − H]^−^ or 135.04406 [caffeoyl − CO_2_ − H]^−^. We tried to explain the reason as following: when the quinic acid compound was ionized, the protons at the 1-carboxyl were easily lost, forming a negative charge center. When the charge migrated to the binding oxygen atom between the quinic acid nucleus and the substituted caffeoyl group, the cleavage could occur, and *m/z* 191.05501 [quinic acid − H]^−^ and 173.04445 [quinic acid – H − H_2_O]^−^ appeared. While when the 1-carboxyl was methylated, the negative charge center will be generated on the hydroxyl group of the substituted caffeoyl group, and after electron transferred, the chemical bond between the substituted caffeoyl group and the quinic acid nucleus would be broken, and a characteristic ion formed by substituted caffeoyl group would strongly occur on the MS/MS spectrum.

## 4. Conclusion

In summary, a method integrating normal-phase chromatography and reverse-phase chromatography/mass spectrometry analysis was established to accomplish the chemical profiling of PI. According to the retention time (*t_R_*) and the exact mass-to-charge ratio (*m/z*), 114 compounds were identified or tentatively identified. Among them, 67 compounds were unambiguously identified by comparing to the standard references. Meanwhile, the rules of MS/MS fragmentation pattern and chromatographic elution order have been generalized by using the reference standard, 47 compounds were tentatively speculated, and 10 of them (peaks **3**, **9**, **15**, **28**, **92**, **97**, **98**, **104**, **106**, and **110**) are potentially novel. An accurate and comprehensive chemical composition profiling of the aerial part of the *P. indica* was realized, which laid a foundation for the quality evaluation of the plant.

## Figures and Tables

**Figure 1 molecules-24-02784-f001:**
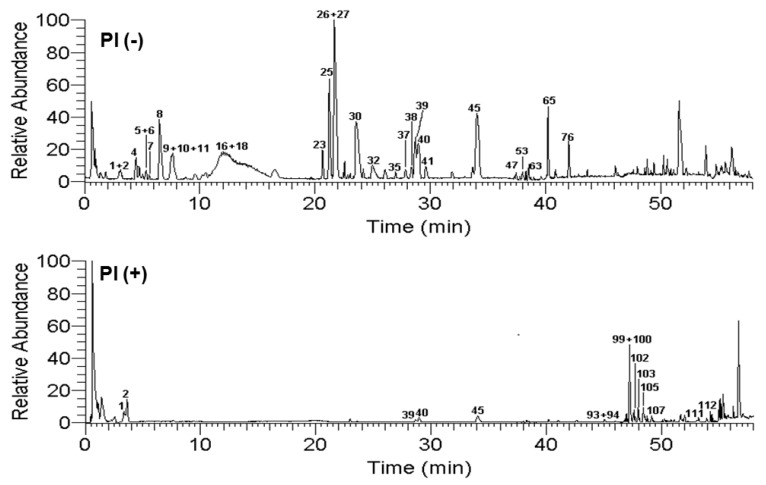
Base peak chromatograms of *P. indica* 70% EtOH extract on BEHC18 column in negative and positive mode.

**Figure 2 molecules-24-02784-f002:**
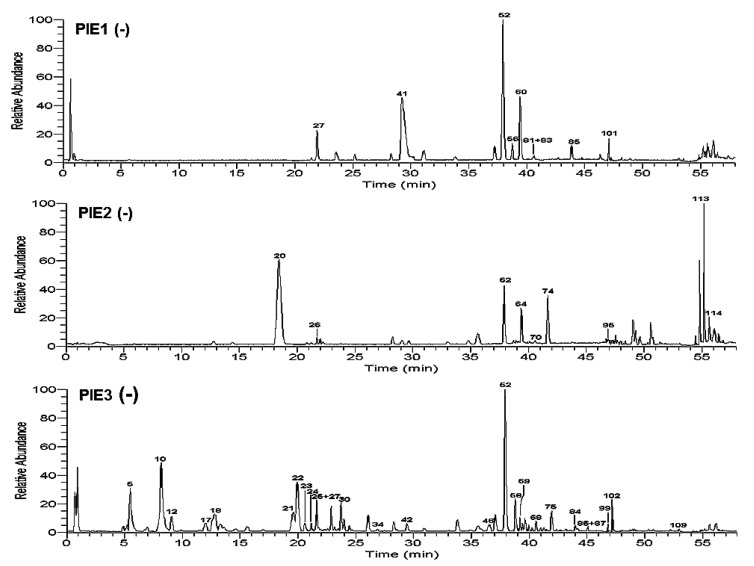
Base peak chromatograms of silica gel fractionation of 95% EtOH eluate of *P. indica* on BEHC18 column in negative mode.

**Figure 3 molecules-24-02784-f003:**
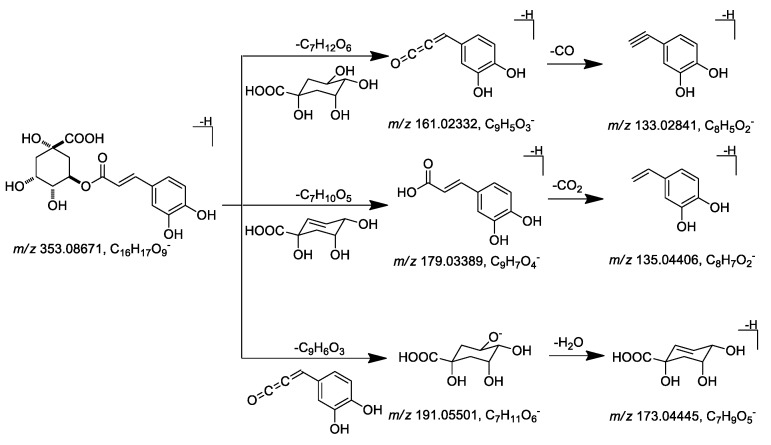
Proposed fragmentation patterns and characteristic ions of caffeoylquinic acid.

**Figure 4 molecules-24-02784-f004:**
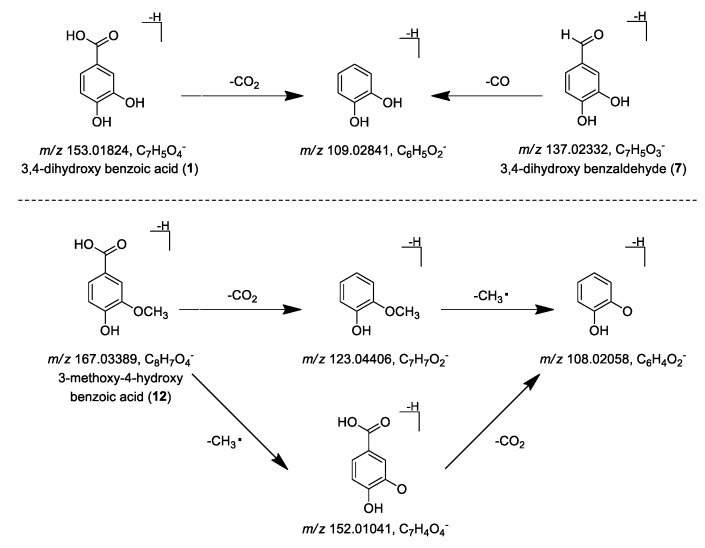
Proposed fragmentation patterns and characteristic ions of 3,4-dihydroxy benzoic acid (**1**), 3,4-dihydroxy benzaldehyde (**7**) and 3-methoxy-4-hydroxy benzoic acid (**12**).

**Figure 5 molecules-24-02784-f005:**
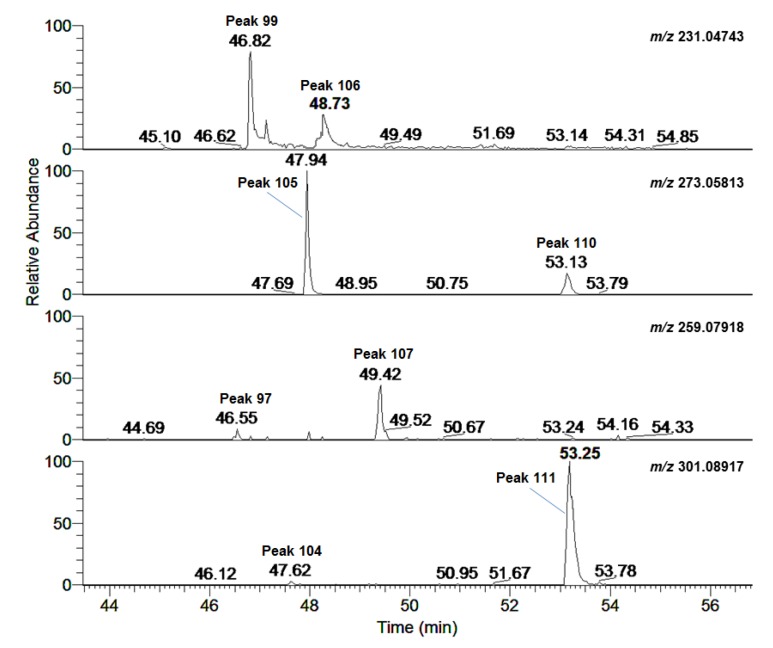
EIC from PIE2 A: EIC of *m/z* 231.04743; B: EIC of *m/z* 273.05813; C: EIC of *m/z* 259.07918; D: EIC of *m/z* 301.08917.

**Table 1 molecules-24-02784-t001:** The qualitative analysis of compounds **1**–**114** by ESI-Q-Orbitrap mass spectrometry (MS).

No.	*t*_R_ (min)	Compound	Formula	Adduct Ions	Theoretic	Measure	Diff (ppm)	Identification Fragment Ions (*m/z*) (Intensity)
**1**	3.17	3,4-Dihydroxy benzoic acid *	C_7_H_6_O_4_	[M − H]^−^	153.01933	153.01914	−1.24	109.02793 (100)
**2**	3.51	Adenosine *	C_10_H_13_N_5_O_4_	[M − H]^−^	266.08948	266.08969	0.79	134.04578 (100)
**3**	4.26	5-TQA	C_16_H_18_O_10_	[M − H]^−^	369.08272	369.08298	0.70	195.02939(16), 191.05483 (100), 133.02803 (82)
**4**	4.97	5-CQA *	C_16_H_18_O_9_	[M − H]^−^	353.08781	353.08826	1.27	191.05497 (100), 179.03378 (54), 135.04367 (62)
**5**	5.27	2,3-Dihydroxy-1-(4-hydroxy-3-methoxyphenyl)-propan-1-one *	C_10_H_12_O_5_	[M − H]^−^	211.06120	211.06069	−2.41	181.04950 (100), 163.03879 (88)
**6**	5.28	*p*-Hydroxybenzoic acid *	C_7_H_6_O_3_	[M − H]^−^	137.02442	137.02309	−0.23	93.03307 (100)
**7**	5.54	3,4-Dihydroxybenzaldehyde *	C_7_H_6_O_3_	[M − H]^−^	137.02442	137.02413	−2.12	109.02798 (14)
**8**	6.69	Methyl 3,4-dihydroxybenzoate	C_8_H_8_O_4_	[M − H]^−^	167.03498	167.03418	−4.79	152.01022 (100), 108.02016 (2)
**9**	7.39	3-TQA	C_16_H_18_O_10_	[M − H]^−^	369.08272	369.08316	−1.19	195.02869 (60), 133.02803 (100)
**10**	8.13	3,4-Dihydroxy-5-methoxybenzaldehyde *	C_8_H_8_O_4_	[M − H]^−^	167.03498	167.03424	−4.43	152.01013 (100), 124.01488 (3)
**11**	8.20	3-CQA *	C_16_H_18_O_9_	[M − H]^−^	353.08781	353.08820	1.10	191.05495 (100), 179.03383 (1), 135.04362 (1)
**12**	8.41	3-Methoxy-4-hydroxybenzoic acid *	C_8_H_8_O_4_	[M − H]^−^	167.03498	167.03430	−4.07	152.01022 (100), 123.04363 (8), 108.02016 (21)
**13**	9.30	Esculetin *	C_9_H_6_O_4_	[M − H]^−^	177.01933	177.01863	−3.95	141.8685 (72), 133.02802 (100)
**14**	9.41	Caffiec acid *	C_9_H_8_O_4_	[M − H]^−^	179.03498	179.03427	−3.97	135.04359 (100)
**15**	10.26	4-TQA	C_16_H_18_O_10_	[M − H]^−^	369.08272	369.08322	1.35	195.02857 (11), 173.04421 (100)
**16**	10.69	4-CQA *	C_16_H_18_O_9_	[M − H]^−^	353.08781	353.08762	−0.54	191.05493 (49), 179.03373 (66), 173.04429 (100), 135.04366 (72)
**17**	12.03	2-Hydroxybenzoic acid or 3-Hydroxybenzoic acid	C_7_H_6_O_3_	[M − H]^−^	137.02442	137.02413	−2.12	93.03311 (100)
**18**	14.27	3-CoQA	C_16_H_18_O_8_	[M − H]^−^	337.09289	337.09384	2.82	191.05482 (100), 163.03868 (9)
**19**	17.63	Vanillin *	C_8_H_8_O_3_	[M − H]^−^	151.04007	151.03996	−0.73	136.01509 (100), 109.02799 (7)
**20**	18.44	4-CoQA	C_16_H_18_O_8_	[M − H]^−^	337.09289	337.09351	1.84	173.04419 (100), 163.03868 (19)
**21**	19.61	5-FQA	C_17_H_20_O_9_	[M − H]^−^	367.10346	367.10382	0.98	193.04926 (1), 161.02310 (100)
**22**	19.94	3-FQA	C_17_H_20_O_9_	[M − H]^−^	367.10346	367.10373	0.74	193.04933 (7), 191.05495 (100)
**23**	20.82	Syringicaldehyde *	C_9_H_10_O_4_	[M − H]^−^	181.05063	181.05087	1.59	166.02563 (100), 151.00235 (86)
**24**	21.24	4-FQA	C_17_H_20_O_9_	[M − H]^−^	367.10346	367.10376	0.82	193.04939 (17), 173.04428 (100)
**25**	21.28	Monosulfonyl substituted flavone	C_15_H_10_O_10_S	[M − H]^−^	380.99219	380.99155	1.68	301.03424 (100), 178.99715 (2), 151.00209 (10)
**26**	21.73	Monosulfonyl substituted flavone	C_15_H_10_O_10_S	[M − H]^−^	380.99219	380.99167	−1.36	301.03586 (100), 178.99789 (25), 151.00278 (65)
**27**	21.80	Disulfonyl substituted flavone	C_15_H_10_O_13_S_2_	[M − H]^−^	460.94900	460.94988	1.91	380.99248 (75), 301.03577 (100), 178.99805 (19), 151.00278 (40), 96.95886 (18)
**28**	22.29	5-CQM	C_17_H_20_O_9_	[M − H]^−^	367.10346	367.10373	0.74	179.03380 (7), 173.04451 (1), 161.02313 (100), 133.02809 (28)
**29**	23.12	3-CQM *	C_17_H_20_O_9_	[M − H]^−^	367.10346	367.10379	0.90	179.03371 (55), 161.02318 (26), 135.04358 (100)
**30**	23.73	Monosulfonyl substituted flavone	C_15_H_10_O_10_S	[M − H]^−^	380.99219	380.99255	0.94	301.03479 (100), 178.99739 (21), 151.00240 (53)
**31**	24.73	Quercetin-3-*O*-β-d-galactopyranoside *	C_21_H_20_O_12_	[M + H]^+^	463.08820	463.08654	−3.58	300.02689 (100), 271.02435 (51), 255.02951 (23), 243.02861 (10)
**32**	25.48	Quercetin-3-*O*-β-d-glucopyranoside *	C_21_H_20_O_12_	[M + H]^+^	463.08820	463.08749	−1.53	300.03 (100), 271.02424 (71), 255.02955 (31), 243.02860 (13)
**33**	26.49	(+)-Isolariciresinol *	C_20_H_24_O_6_	[M − H]^−^	359.15001	359.14966	−0.97	ND
**34**	26.87	*trans*-Ferulic acid *	C_10_H_10_O_4_	[M − H]^−^	193.05063	193.05017	−2.38	161.02310 (28), 133.02797 (19)
**35**	27.11	1,4-DiCQA *	C_25_H_24_O_12_	[M − H]^−^	515.11950	515.11865	−1.65	353.08719 (60), 191.05486 (46), 179.03369 (69), 173.04434 (100), 135.04370 (77)
**36**	27.63	Kaempferol-*O*-galactopyranoside	C_21_H_20_O_11_	[M − H]^−^	447.09328	447.09430	2.28	284.03174 (84), 255.02882 (100), 227.03365 (82)
**37**	27.99	1,5-DiCQA *	C_25_H_24_O_12_	[M − H]^−^	515.11950	515.12067	2.27	353.08688 (17), 191.05472 (100), 179.03371 (5), 161.02297 (5), 135.04356 (6)
**38**	28.22	1,3-DiCQA *	C_25_H_24_O_12_	[M − H]^−^	515.11950	515.11871	−1.53	353.08694 (12), 191.05495 (100), 179.03365 (4), 161.02303 (5), 135.04370 (6)
**39**	28.68	4,5-DiCQA *	C_25_H_24_O_12_	[M − H]^−^	515.11950	515.12006	1.09	353.08713 (59), 191.05493 (53), 179.03375 (84), 173.04420 (100), 161.02312 (37), 155.03355 (14), 135.04362 (85)
**40**	28.89	3,5-DiCQA *	C_25_H_24_O_12_	[M − H]^−^	515.11950	515.11888	−1.20	353.08697 (57), 191.05480 (100), 179.03369 (51), 173.04446 (5), 135.04363 (54)
**41**	29.23	Monosulfonyl substituted methoxyflavone	C_16_H_12_O_10_S	[M − H]^−^	395.00784	395.00778	−0.15	315.05048 (100), 300.02689 (41), 271.02438 (26), 255.02913 (13), 243.02908 (6)
**42**	29.63	Kaempferol 3-*O*-β-d-glucopyranoside *	C_21_H_20_O_11_	[M − H]^−^	447.09328	447.09423	2.12	284.03159 (79), 255.02875 (100), 227.03368 (87)
**43**	30.39	Cynaroside *	C_21_H_20_O_11_	[M − H]^−^	447.09328	447.09406	1.74	ND
**44**	31.20	5,7,3',4'-Tetrahydroxy-3-methoxy flavonol-3'-*O*-β-d-glucopyranoside *	C_22_H_22_O_12_	[M − H]^−^	477.10385	477.10520	2.83	ND
**45**	33.94	3,4-DiCQA *	C_25_H_24_O_12_	[M − H]^−^	515.11950	515.12002	1.01	353.08701 (71), 191.05498 (37), 179.03368 (64), 173.04430 (100), 135.04367 (62)
**46**	36.21	1-C-5-F-QA or 1-F-5-C-QA	C_26_H_26_O_12_	[M − H]^−^	529.13515	529.13550	0.66	367.10258 (21), 353.08741 (22), 349.09223 (20), 335.07694 (25), 193.04936 (93), 179.03373 (66), 173.04420 (83), 161.02299 (42), 155.03351 (39), 135.04369 (68), 134.03582 (100)
**47**	36.63	1,3,5-TriCQA *	C_34_H_30_O_15_	[M − H]^−^	677.15119	677.15155	0.53	515.11898 (5), 497.10879 (10), 353.08703 (50), 335.07655 (13), 191.05478 (100), 179.03362 (70), 173.04419 (11), 161.02305 (23), 135.04358 (64)
**48**	37.15	1-C-3-FQA	C_26_H_26_O_12_	[M − H]^−^	529.13515	529.13581	1.25	367.10327 (43), 191.05527 (100), 173.04451 (54), 161.02300 (12), 134.03579 (18)
**49**	37.30	3-F-5-CQA	C_26_H_26_O_12_	[M − H]^−^	529.13515	529.13574	1.12	367.10309 (42), 193.04980 (100), 173.04443 (21), 134.03607 (81)
**50**	37.49	1,4-DiCQM	C_26_H_26_O_12_	[M − H]^−^	529.13515	529.13416	−1.87	367.10264 (24), 353.08711 (29), 191.05522 (100), 179.03364 (38), 161.02306 (40), 135.04353 (50)
**51**	37.77	4,5-DiCQM *	C_26_H_26_O_12_	[M − H]^−^	529.13515	529.13501	−0.26	367.10242 (15), 179.03372 (4), 161.02308 (100), 133.02805 (28)
**52**	37.91	Ethyl caffeate *	C_11_H_12_O_4_	[M − H]^−^	207.06628	207.06556	−3.48	179.03377 (27), 159.85886 (41), 135.04372 (44), 127.86897 (35), 103.91862 (14)
**53**	38.22	1,3,4-TriCQA *	C_34_H_30_O_15_	[M − H]^−^	677.15119	677.15070	−0.72	515.11847 (34), 497.10794 (18), 353.08694 (19), 335.07701 (10), 191.05501 (46), 179.03377 (100), 173.04422 (83), 161.02305 (78), 135.04367 (83)
**54**	38.32	Quercetin *	C_15_H_10_O_7_	[M − H]^−^	301.03538	301.03571	1.10	272.02643 (6), 178.99734 (24), 151.00224 (70), 121.02786 (19), 107.01236 (19)
**55**	38.39	3,5-DiCQM *	C_26_H_26_O_12_	[M − H]^−^	529.13515	529.13519	0.08	367.10275 (22), 179.03320 (31), 161.02304 (100), 135.04358 (45)
**56**	38.45	Triethyl citrate *	C_12_H_20_O_7_	[M − H]^−^	275.11362	275.11278	−3.05	ND
**57**	38.49	Luteolin *	C_15_H_10_O_6_	[M − H]^−^	285.04046	285.04114	2.39	151.00223 (10), 133.02827 (32)
**58**	38.79	CFQM-1	C_27_H_28_O_12_	[M − H]^−^	543.15080	543.14954	−2.32	367.10281 (13), 349.09195 (75), 193.04941 (75), 173.04420 (95), 161.02319 (9), 155.03357 (42), 134.03580 (100)
**59**	39.04	(–)-(7*S*,7'*S*,8*R*,8'*R*)-4,4'-Dihydroxy-3,3',5,5'-pentamethoxy-7,9':7',9-diepoxylignane *	C_22_H_26_O_8_	[M − H]^−^	415.15748	415.15757	0.22	ND
**60**	39.39	CFQM-2	C_27_H_28_O_12_	[M − H]^−^	543.15080	543.15009	−1.31	367.10251 (25), 349.09210 (18), 193.04941 (100), 161.02301 (49), 134.03583 (18)
**61**	39.49	3,4-DiCQM *	C_26_H_26_O_12_	[M − H]^−^	529.13515	529.13562	0.89	367.10202 (17), 179.03372 (69), 161.02311 (63), 135.04359 (100)
**62**	39.59	1,5-DiCQE/4,5-DiCQE	C_27_H_28_O_12_	[M − H]^−^	543.15080	543.15125	0.83	381.11993 (12), 161.02313 (100), 133.02809 (28)
**63**	39.85	1,4,5-TriCQA	C_34_H_30_O_15_	[M − H]^−^	677.15119	677.15173	0.80	515.11743 (15), 353.08701 (55), 191.05486 (43), 179.03371 (89), 173.04424 (100), 161.02301 (33), 135.043 64(80)
**64**	39.89	Methyl 9-hydroxynonanoate *	C_10_H_20_O_3_	[M − H]^−^	187.13397	187.13403	0.32	141.12700 (100)
**65**	40.13	3,4,5-TriCQA *	C_34_H_30_O_15_	[M − H]^−^	677.15119	677.15179	0.89	515.11885 (22), 353.08701 (33), 191.05479 (61), 179.03371 (96), 173.04419 (100), 161.02303 (43), 135.04364 (84)
**66**	40.13	3,5-DiCQE *	C_27_H_28_O_12_	[M − H]^−^	543.15080	543.15094	0.26	381.11856 (20), 179.03387 (33), 161.02316 (100), 133.02808 (32)
**67**	40.33	CFQM-3	C_27_H_28_O_12_	[M − H]^−^	543.15080	543.15118	0.70	367.10272 (23), 349.09177 (33), 193.04958 (34), 179.03358 (60), 173.04431 (17), 161.02312 (80), 135.04375 (100), 134.03601 (39), 133.02805 (26)
**68**	40.34	5,6,4'-Trihydroxy-3,7-dimethoxyflavone *	C_17_H_14_O_7_	[M − H]^−^	329.06667	329.06669	0.06	314.04276 (100), 299.01917 (94), 271.02429 (33), 243.02908 (13)
**69**	40.84	CFQM-4	C_27_H_28_O_12_	[M − H]^−^	543.15080	543.15088	0.15	193.04944 (13), 161.02306 (100), 134.03596 (20), 133.02803 (34)
**70**	41.10	TriMethoxyflavone	C_18_H_16_O_8_	[M − H]^−^	359.07724	359.07785	1.70	344.05280 (97), 329.02954 (100), 314.00613 (7), 301.03461 (12), 286.01123 (27), 258.01624 (16)
**71**	41.24	Kaempferol *	C_15_H_10_O_6_	[M−H]^−^	285.04046	285.03998	−1.68	151.00218 (2)
**72**	41.39	CFQM-5	C_27_H_28_O_12_	[M−H]^−^	543.15080	543.15106	0.48	349.09198 (52), 193.04942 (70), 179.03371 (23), 161.02310 (100), 135.04375 (40), 134.03590 (91), 133.02800 (26)
**73**	41.61	3,4-DiCQE *	C_27_H_28_O_12_	[M − H]^−^	543.15080	543.15094	0.26	381.11804 (16), 161.02304 (75), 179.03371 (68), 135.04361 (100)
**74**	41.68	*threo*-2,3-Bis(4-hydroxy-3-methoxyphenyl)-3-ethoxypropan-1-ol *	C_19_H_24_O_6_	[M − H]^−^	347.15001	347.14891	−3.17	161.02312 (9), 135.04370 (16)
**75**	41.91	9,12,13-Trihydroxyoctadeca-10(*E*),15(*Z*)-dienoic acid *	C_18_H_32_O_5_	[M − H]^−^	327.21770	327.21742	−0.86	229.14348 (16), 211.13269 (27), 171.10144 (23)
**76**	41.98	1,3,4,5-TetraCQA *	C_43_H_36_O_18_	[M − H]^−^	839.18289	839.18365	0.91	659.13635 (6), 515.11799 (20), 353.08657 (10), 335.07645 (8), 191.05495 (41), 179.03377 (100), 173.04430 (73), 161.02301 (61), 135.04375 (91)
**77**	42.08	Isorhamnetin *	C_16_H_12_O_7_	[M − H]^−^	315.05103	315.05127	0.76	300.02676 (88), 271.02344 (6), 164.01016 (7), 151.00218 (22)
**78**	42.10	CFQM-6	C_27_H_28_O_12_	[M − H]^−^	543.15080	543.15131	0.94	349.09146 (16), 193.04938 (50), 161.02304 (100), 135.04370 (20), 134.03587 (77), 133.02797 (34)
**79**	42.10	1,3,4-CCFQA	C_35_H_32_O_15_	[M − H]^−^	691.16684	691.16785	1.46	529.13403 (10), 367.10284 (20), 349.09241 (9), 193.04938 (100), 179.03378 (73), 161.02312 (58), 135.04370 (44), 133.02805 (84)
**80**	42.32	1,3,4-TriCQM	C_35_H_32_O_15_	[M − H]^−^	691.16684	691.16760	1.10	529.1361 (17), 515.11920 (22), 353.08786 (42), 367.10275 (10), 335.07806 (8), 191.05534 (77), 179.03372 (89), 173.04465 (100), 161.02309 (47), 155.03362 (17), 135.04393 (85)
**81**	42.55	*erythro*-2,3-Bis(4-hydroxy-3-methoxyphenyl)-3-ethoxypropan-1-ol *	C_19_H_24_O_6_	[M − H]^−^	347.15001	347.14891	−3.17	179.03271 (81), 135.04364 (100)
**82**	42.61	1,4,5-CCFQA	C_35_H_32_O_15_	[M − H]^−^	691.16684	691.16772	1.27	529.13336 (25), 367.10263 (25), 335.07651 (5), 193.04953 (16), 179.03380 (20), 173.04433 (100), 161.02312 (22), 155.03381 (11), 135.04378 (21), 133.02805 (15)
**83**	43.13	Pinellic acid *	C_18_H_34_O_5_	[M − H]^−^	329.23335	329.23300	−1.06	229.14352 (8), 211.13301 (17), 171.10135 (32)
**84**	43.58	Trihydroxy-dimethoxyflavone	C_17_H_14_O_7_	[M − H]^−^	329.06667	329.06705	1.15	ND
**85**	43.87	Caryolane-1,9β-diol *	C_15_H_26_O_2_	[M − H_2_O + H]^+^	221.18999	221.18954	−2.03	ND
**86**	44.19	Chrysosplenol C *	C_18_H_16_O_8_	[M − H]^−^	359.07724	359.07773	1.36	344.05292 (94), 329.02963 (100), 286.01129 (71)
**87**	44.30	Centaureidin *	C_18_H_16_O_8_	[M − H]^−^	359.07724	359.07791	1.87	344.05289 (89), 329.02954 (89), 344.05289 (89), 329.02954 (89), 314.00644 (40), 301.03482 (17), 286.01126 (97)
**88**	44.40	CCCFQA-1	C_44_H_38_O_18_	[M − H]^−^	853.19854	853.19971	1.37	645.15802 (10), 529.13397 (78), 335.07648 (15), 193.04938 (94), 179.03372 (86), 173.04422 (75), 161.02307 (100), 155.03362 (34), 135.04372 (94)
**89**	44.46	CCFFQA-1	C_45_H_40_O_18_	[M − H]^−^	867.21419	867.21277	−1.64	193.04977 (5), 161.02315 (100), 135.04369 (11), 133.02803 (47)
**90**	44.77	CCCFQA-2	C_44_H_38_O_18_	[M − H]^−^	853.19854	853.19946	1.08	645.16498 (3), 529.13361 (29), 367.10287 (9), 193.04933 (20), 179.03880 (21), 173.04419 (100), 161.02298 (27), 135.04361 (26)
**91**	44.93	3,4,5-TriCQM *	C_35_H_32_O_15_	[M − H]^−^	691.16684	691.16754	1.01	179.03409 (29), 161.02346 (100), 135.04391 (45), 133.02799 (31)
**92**	45.68	1,3,4,5-TetraCQE	C_45_H_40_O_18_	[M − H]^−^	867.21419	867.21356	−0.73	355.08116 (26), 179.03372 (23), 161.02294 (100), 135.04371 (47), 133.02806 (43)
**93**	45.88	Fraxinellone *	C_14_H_16_O_3_	[M + H]^+^	233.11722	233.11784	2.66	215.10712 (100), 187.11209 (92)
**94**	46.07	valenc-1(10)-ene-8,11-diol *	C_15_H_26_O_2_	[M − H_2_O + H]^+^	221.18999	221.18974	−1.13	203.17982 (100), 161.13272 (21), 147.11707 (45), 133.10150 (30), 119.08596 (81), 109.10172 (35), 107.08604 (35), 95.08617 (56)
**95**	46.39	Trihydroxy-dimethoxyflavone	C_17_H_14_O_7_	[M − H]^−^	329.06667	329.06699	0.97	ND
**96**	46.12	CCFFQA-2	C_45_H_40_O_18_	[M − H]^−^	867.21419	867.21252	−1.93	161.02298 (44), 135.04358 (17), 133.02797 (15)
**97**	46.55	3''-Ethoxyl-(3''*R*)-pluthiophenol	C_15_H_14_O_2_S	[M + H]^+^	259.07873	259.07944	2.74	213.03680 (100), 199.02121 (55), 185.04228 (53), 173.00558 (78)
**98**	46.57	3,4,5-TriCQE	C_36_H_34_O_15_	[M − H]^−^	705.18249	705.18292	0.61	543.15044 (8), 367.10269 (2), 349.09255 (4), 179.03368 (47), 161.02313 (100), 135.04359 (80)
**99**	46.82	(3''*R*)-pluthiophenol *	C_13_H_10_O_2_S	[M + H]^+^	231.04743	231.04755	0.52	213.04740 (36), 200.02346 (17), 199.02358 (53)
**100**	46.89	Clovane-2α,9β-diol *	C_15_H_26_O_2_	[M − H_2_O + H]^+^	221.18999	221.19037	1.72	203.17920 (100), 161.13242 (11), 147.11676 (33), 121.10132 (20), 109.10140 (31), 107.08585 (31), 95.08591 (58)
**101**	47.08	Casticin *	C_19_H_18_O_8_	[M − H]^−^	373.09289	373.09366	2.06	358.06879 (77), 343.04517 (100), 328.02164 (18), 312.99838 (13), 300.02676 (33), 285.00342 (34), 257.00858 (33)
**102**	47.17	(8*R*,9*R*)-Isocaryolane-8,9-diol *	C_15_H_26_O_2_	[M − H_2_O + H]^+^	221.18999	221.18962	−1.67	ND
**103**	47.53	Sesquiterpene	C_15_H_26_O_2_	[M − H_2_O + H]^+^	221.18999	221.18941	−2.62	ND
**104**	47.62	3''-Ethoxyl-(3''*R*)-pluthiophenol-4''-acetate	C_17_H_16_O_3_S	[M + H]^+^	301.08929	301.08945	0.53	ND
**105**	47.96	(3''*R*)-Pluthiophenol-4''-acetate *	C_15_H_12_O_3_S	[M + H]^+^	273.05799	273.05813	0.51	231.04753 (40), 213.03694 (76), 184.03423 (100)
**106**	48.73	(3''*S*)-pluthiophenol	C_13_H_10_O_2_S	[M + H]^+^	231.04743	231.04758	0.65	ND
**107**	49.42	3''-Ethoxyl-(3''*S*)-pluthiophenol *	C_15_H_14_O_2_S	[M + H]^+^	259.07873	259.07918	1.74	213.03680 (100), 199.02121 (42), 185.04228 (43), 173.00558 (73)
**108**	49.62	Sesquiterpene	C_15_H_26_O_2_	[M − H_2_O + H]^+^	221.18999	221.18947	−2.35	ND
**109**	53.00	Dibutylphthalate *	C_16_H_22_O_4_	[M − H]^−^	277.15899	277.15901	0.07	ND
**110**	53.13	(3''*S*)-Pluthiophenol-4''-acetate	C_15_H_12_O_3_S	[M + H]^+^	273.05799	273.05838	1.43	231.04761 (40), 213.03699 (75), 184.03421 (100)
**111**	53.28	3''-Ethoxyl-(3''*S*)-pluthiophenol-4''-acetate *	C_17_H_16_O_3_S	[M + H]^+^	301.08929	301.08917	−0.40	213.03697 (100), 185.04243 (45), 184.03429 (48), 173.00571 (8)
**112**	54.98	*trans*-Coniferyl aldehyde *	C_10_H_10_O_3_	[M + H]^+^	179.07027	179.07063	2.01	ND
**113**	55.18	(+)-9'-isovaleryllariciresinol *	C_25_H_32_O_7_	[M − H]^−^	443.20753	443.20905	3.43	ND
**114**	55.94	Stigmasterol *	C_29_H_48_O	[M − H]^−^	411.36324	411.36307	−0.41	ND

* The compounds unambiguously identified with the reference standards comparison; CoQA: coumaroyl quinic acid; CQA: caffeoyl quinic acid; CQE: ethyl caffeoyl quinate; CQM: methyl caffeoyl quinate; CFQM: methyl caffeoyl-ferulyl-quinate; CCFQA: dicaffeoyl-ferulyl-quinic acid; CCCFQA: tricaffeoyl-ferulyl-quinic acid; CCFFQA: dicaffeoyl-diferulyl-quinic acid; FQA: feruloyl quinic acid; TQA: 3,4,5-trihydroxycinnamoyl quinic acid. Although the compounds **33**, **43**, **44**, **56**, **59**, **85**, **102**, **109**, and **112**−**114** were identified by the comparison of *t*_R_ with reference standards, while the contents of them were too low to detect their fragments ions, thus they were stated as ND.

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
