# Peer review of "Comprehensive Chemical Profiling in the Ethanol Extract of Pluchea indica Aerial Parts by Liquid Chromatography/Mass Spectrometry Analysis of Its Silica Gel Column Chromatography Fractions"

_molecules, 2019, doi:10.3390/molecules24152784_

Round 1
Reviewer 1 Report
Although the paper is well presented and provides an extensive and comprehensive chemical Profiling of Pluchea indica aerial parts using well-known separation and identification analytical techniques, I have some minor concerns and comments that need to be addressed.
1. Introduction
This section needs more improvements. For instance, information regarding the cytotoxicity and safety profile of P. indica is missed, where the plant is being used as a medicinal and food plant. Moreover, the pharmacological/biological effects that are induced by this plant are also unspecified. These points should be addressed in this section.
2. Experimental
I would recommend the authors to provide the full description of all reported abbreviations in sequence within the paper.
Table1, although it was stated with marked (*) that some compounds unambiguously identified with the reference standards comparison, some of those compounds with no identified fragments ions were reported (as stated ND) and some others with identified fragments ions (even with a comparison with standard compounds). Please, provide an explanation for such matter and mention it below the table.
Please, provide the UHPLC chromatogram of all separated compounds.
In general, I would recommend the authors to pay attention to grammatical and typing errors.
Reviewer 2 Report
The authors report on a pure and simple work of analytical chemistry, applied to a wide chemical profile of fractions from aerial parts of Pluchea indica extract, also using a large number of reference compounds. The title is misleading and must be modified as "Comprehensive ......aerial parts by liquid chromatography/ mass spectrometry analysis of silica gel column chromatography fractions from ethanol extract”, or similarly.
At the beginning of Results and discussion the authors must introduce a short definition of orthogonal chromatography, adopted as a method to select the best stationary phase in LC-MS analysis of the fractions.
There is too much use of acronyms, e.g. they can be rplaced by the full names in the figure captions and in the titles of paragraphs (e.g. PI, PIE).
At page 5, line 182, TCM must be replaced by the full names.
The title of paragraph 3.1 is wrongly reported in the caption of figure 2.
Reviewer 3 Report
Expand the introduction with publications of the activity of the Pluchea indica based on their anti-obesity and antioxidant activities:
1) Kittipot Sirichaiwetchakoon, Gordon Matthew Lowe, Kanjana Thumanu, and Griangsak Eumkeb, “The Effect of Pluchea indica (L.) Less. Tea on Adipogenesis in 3T3-L1 Adipocytes and Lipase Activity,” Evidence-Based Complementary and Alternative Medicine, vol. 2018, Article ID 4108787, 13 pages, 2018. https://doi.org/10.1155/2018/4108787.
2) Hafeedza Abdul Rahman, Nazamid Saari, Faridah Abas, Amin Ismail, Muhammad Waseem Mumtaz & Azizah Abdul Hamid (2017). Anti-obesity and antioxidant activities of selected medicinal plants and phytochemical profiling of bioactive compounds,International Journal of Food Properties, 20:11, 2616-2629, DOI: 10.1080/10942912.2016.1247098
